# Different Growth and Sporulation Responses to Temperature Gradient among Obligate Apomictic Strains of *Ulva prolifera*

**DOI:** 10.3390/plants10112256

**Published:** 2021-10-22

**Authors:** Yoichi Sato, Yutaro Kinoshita, Miho Mogamiya, Eri Inomata, Masakazu Hoshino, Masanori Hiraoka

**Affiliations:** 1Bio-Resources Business Development Division, Riken Food Co., Ltd., Miyagi 985-0844, Japan; yuu_kinoshita@rikenfood.co.jp (Y.K.); mih_mogamiya@rikenfood.co.jp (M.M.); eri_inomata@rikenfood.co.jp (E.I.); 2Nishina Center for Accelerator-Based Science, RIKEN, Saitama 351-0198, Japan; 3Usa Marine Biological Institute, Kochi University, Kochi 781-1164, Japan; 4Department of Algal Development and Evolution, Max Planck Institute for Developmental Biology, Max-Planck-Ring 5, 72076 Tübingen, Germany; mhoshino.sci@gmail.com

**Keywords:** macroalga, *Ulva prolifera*, obligate asexual strain, relative growth rate, sporulation, land-based cultivation, germling cluster method

## Abstract

The green macroalga *Ulva prolifera* has a number of variants, some of which are asexual (independent from sexual variants). Although it has been harvested for food, the yield is decreasing. To meet market demand, developing elite cultivars is required. The present study investigated the genetic stability of asexual variants, genotype (*hsp90* gene sequences) and phenotype variations across a temperature gradient (10–30 °C) in an apomictic population. Asexual variants were collected from six localities in Japan and were isolated as an unialgal strain. The *hsp90* gene sequences of six strains were different and each strain included multiple distinct alleles, suggesting that the strains were diploid and heterozygous. The responses of growth and sporulation versus temperature differed among strains. Differences in thermosensitivity among strains could be interpreted as the result of evolution and processes of adaptation to site-specific environmental conditions. Although carbon content did not differ among strains and cultivation temperatures, nitrogen content tended to increase at higher temperatures and there were differences among strains. A wide variety of asexual variants stably reproducing clonally would be advantageous in selecting elite cultivars for long-term cultivation. Using asexual variants as available resources for elite cultivars provides potential support for increasing the productivity of *U. prolifera*.

## 1. Introduction

The green macroalga *Ulva prolifera* O.F. Müller, 1778 (Class Ulvophyceae) is an example of an alga showing isomorphic alternation of generations, with sporophytes and gametophytes that are morphologically indistinguishable. In the life cycle of *Ulva*, gametophytes of two mating types release biflagellate gametes with positive phototaxis, and the zygote develops into the sporophytic phase [1]. Sporophytes release quadriflagellate meiospores through meiosis, which develop into genetically separate gametophytes [2]. In addition to the sexual life cycle, several *Ulva* species, including *U. prolifera*, are known to have two types of obligate asexual life cycles without sexual reproduction via meiosis and conjugation, reproducing through biflagellate or quadriflagellate diploid zoids specialized for asexual development, these zoids have negative phototaxis [3]. These asexual zoids were termed “zoosporoids” [4,5]. The quadriflagellate zoosporoids of obligate asexual life history were a length of <10 µm; these were smaller in size than quadriflagellate meiospores of sexual life history (>11 µm length). On the other hand, biflagellate zoosporoids of obligate asexual life history were a length of 8–9 µm; these were distinguished from biflagellate games (6–7 µm length) [6]. These asexual variants are regarded as diploid thalli because the amount of DNA in the cells of asexual thalli is similar to that in the cells of the sporophytic thallus [6,7]. A recent study conducting genome and transcriptome analyses of *U. prolifera* suggests that the asexual thalli originally evolved via apomeiosis in sporophytic thalli [8]. Another study has revealed that the two asexual variants of *U. prolifera* show high levels of heterozygosity in the *hsp90* gene, probably as a result of hybridization among genetically distinct gametophytes [9]. These findings indicate that evolution of the asexual variants of *U. prolifera* has been independently repeated from various genetically distinct sexual populations. Therefore, although asexual *U. prolifera* variants show obligate clonal reproduction, they may include a wide range of genotypes with various physiological characteristics.

The industrial use of *U. prolifera* has a long history, being harvested and used as food in Japan since the 10th century A.D. [10]. Since the 1990s, the artificial seedling method has been developed and the cultivation in estuaries and brackish water areas has been carried out, mainly around Shikoku Island, the 4th largest island of the Japanese Archipelago [11]. However, both cultivated and natural harvest yields have declined markedly due to various environmental factors, particularly salinity and temperature (e.g., [12]). To meet market demand, the “germling cluster” (GC) method was developed as a novel way to raise seedlings and land-based cultivation on an industrial scale began in Japan early this century [13]. Globally, the land-based cultivation of *Ulva* species is focused on biomass production in view of their rapid growth and lack of any requirements for freshwater resources or particular soils for cultivation [14,15,16]. However, it is important to develop a stable cultivation technique in order to achieve sustainable industrial production.

Developing an elite cultivar of *Ulva* species is an essential factor for successful land-based cultivation on a commercial scale, with, in particular, a requirement for a superior growth rate [17]. Since growth rates can vary substantially between cultivars of *Ulva*, both from within the same species [18,19,20] and among different *Ulva* species [21], it is important to compare and select from a wide variety of species and cultivars for optimal biomass production [16]. Asexual variants are particularly appropriate for the selection of elite cultivars because their genetic characteristics can be preserved even across many generations, which is very different from the F_1_ hybridization techniques used with terrestrial crops. Understanding the growth characteristics of asexual variants with respect to environmental differences is also necessary in order to be able to select elite cultivars appropriate to each cultivation area. Additionally, *U. prolifera* thalli often produce and release zoids apically and become shorter in length at 20 °C and over [22]. Therefore, cultivars continuing vegetative growth without sporulation at 20 °C and over are appropriate for land-based cultivation. In short, to select elite cultivars it is important to verify the growth characteristics of different variants not only to optimize growth but also with regard to temperature effects on maturation. However, there have been few studies so far concerning the phenotypic differentiation of *U. prolifera* asexual variants.

The aim of the present study, therefore, is to identify the physiological characteristics for growth and sporulation responses of asexual variants of *U. prolifera* among recognized strains. Asexual thalli were collected from six localities in Japan and compared for their growth rate, carbon and nitrogen contents, and sporulation responses across a range of temperatures using the GC method for seedling production and cultivation to apply the industrial aquaculture perspective.

## 2. Results

### 2.1. Molecular Analysis

The *hsp90* gene sequences of the six strains showed the presence of alternative bases at some positions, indicating the presence of different alleles (Figure 1) and suggesting that these strains are heterozygous and diploid.

### 2.2. Growth Rate and Sporulation at Different Temperatures

The relative growth rate (RGR) at different cultivation temperatures differed among strains. The mean RGR of Strain 1 varied narrowly over the range 0.3–0.4 and was not significantly influenced by temperature (*p* = 0.298, Figure 2, Strain 1), and the values of the maximum were about 1.3 times those of the minimum. However, those of other localities varied among temperature and there were significantly differences by post-hoc tests (*p* < 0.01, Figure 2, Strain 2–6). Although no clear peaks of RGR of Strain 2 and 4–6 were detected, maximum values were detected at 20–30 °C (Figure 2, Strains 2 and 4–6). The maximum values were 2.2–2.9 times those of the minimum. However, RGRs for Strain 3 indicated a clear peak with a mean of 0.55 at 20 °C, which is 3 times faster than the value at 10 °C (Figure 2). The RGRs of Strains 1 and 3 were significantly higher than those of other strains at 10 °C (*p* < 0.05) and 20 °C (*p* < 0.01), respectively (Appendix A). Throughout the cultivation period, no sporulating cells were detected in any of the thalli incubated at 10 or 15 °C (Figure 3). However, from the 2nd day of cultures at 20 °C or above, sporulating cells were already present in Strain 4 thalli (Figure 3), and in the thalli of Strains 1–3, and 5 from the 4th day of culture (Figure 3). In Strains 3–5, sporulating cells occupied more than half the total thallus area at 30 °C (data not shown). Strain 6 thalli showed no evidence of sporulation in cultures incubated below 25 °C, sporulation only began from the 8th day of culture (Figure 3) and was limited to only the tip of the thallus (data not shown).

### 2.3. Carbon and Nitrogen Content at Different Temperatures

The carbon content of thalli for all six strains ranged between 0.337 ± 0.005 and 0.396 ± 0.001 mg mg^−1^, with no obvious peak at any particular incubation temperature, although differences detected among incubation temperatures were significant for Strains 1–3 (Figure 4).

Nitrogen content ranged from 0.035 ± 0.001 (at 10 °C) to 0.055 ± 0.001 mg mg^−1^ (at 30 °C), tending to increase with increasing temperature, for all except Strain 6 (Figure 5). In the latter, no significant differences were detected among different temperature incubations, the values ranging between 0.037 ± 0.001 and 0.0433 ± 0.002 mg mg^−1^ (Figure 5).

## 3. Discussion

The optimum temperature for growth of *U. prolifera* is known to vary according to sampling locality (strain) within the range 15–25 °C (reviewed by [23]), and the thalli generally mature and release zoids at 20 °C or higher [22]. However, previous studies did not identify the generation or type of life cycle of the specimens used in culture experiments. The present study revealed that different strains of asexual thalli of *U. prolifera* had differences in thermosensitivity of growth and sporulation, and that this may be connected with the presence of different *hsp90* genotypes among these different strains. The growth rate temperature optima range within 20–25 °C, which is similar to that reported in previous studies [23]. It is noteworthy that for most strains the growth rate showed significant thermosensitivity. However, the growth rate of Strain 1 was not significantly influenced by temperature, suggesting that it can maintain growth even at lower temperatures. Strain 1 is from the northern Pacific coast of Hokkaido, which is close to the northern limit of distribution of *U. prolifera* [24], so it might be expected to have a lower temperature tolerance than other strains. With regard to sporulation, only Strain 6 thalli did not sporulate at 20 °C within 12 days, demonstrating a clearly different thermosensitivity from other strains, with a greater tolerance to high temperatures for maintaining vegetative growth.

It might be expected that the differences observed in growth rates and temperature tolerance among strains are connected with the environmental characteristics of the locality from which they were obtained. However, the results of the present study indicate that strains from neighboring localities have clearly different thermosensitivity; for instance, the localities of Strains 2 and 3 are only 5 km apart. Considering the genotypic differences among strains of *hsp90*, the phenotypic differences would be caused by genetic background. This suggests that the characteristics of each strain are not necessarily closely adapted to the environmental conditions at the locality in which it is found, and it is considered that the phenotypic diversity of this alga is itself high and not dependent upon site-specific environmental conditions. Hiraoka and Higa (2016) [3] proposed that *U. prolifera* had evolved from an ancestral marine species to become a true estuarine species: firstly, the sexual population adapted to low salinity conditions, and then a number of different asexual generations arose from genetically variable sexual ancestors, with natural selection finally producing an array of specialized asexual thallus genotypes that efficiently occupy the estuarine habitat. The variable thermosensitivity of asexual variants among localities could be interpreted as the result of the evolution and adaptation processes of this alga. The variation of *hsp90* genotypes observed among the six strains may be a manifestation of the phenotypic differentiation among them. Distinguishing among the genotypes affecting the phenotypes requires further study.

Measurements of the net photosynthetic rate and RGRs of *U. prolifera* collected from green tides in China have peak at 18–22 °C with a marked decline at 26 °C [25]. However, in the present study, carbon content was not observed to vary across different culture temperatures for all strains. This demonstrates the potential for stable carbon fixation among strains of *U. prolifera* regardless of temperature fluctuations, implying also a potential for CO_2_ mitigation by *U. prolifera* which could be calculated from yield data.

Nitrogen content, however, varied with temperature for all strains except Strain 6, despite the presence of nutrients sufficient for culture conditions. In previous studies, the nitrogen content of *U. prolifera* collected from eutrophic areas of the Yellow Sea, was reported to be 3.6% [26]. However, values of 3–4% were found in wild-collected thalli from the same locality as Strain 6, where the dissolved inorganic nitrogen concentration measured was found to be insufficient for optimum growth [27]. In the present study, nitrogen content was in the range 2.8–5.1% across different incubation temperatures, suggesting that the assimilation capacity for nitrogen is influenced by temperature. Raven and Geider (1988) previously reported that temperature influences the nutrient-uptake rates via *Q*_10_ effects on algal metabolism [28]. The nitrogen content might therefore reflect the physiological response to differences in temperature which was a variable among the strains in the present study.

Many green algae show rapid nutrient uptake rates, contributing to the removal of excess nutrients in the water column [29,30]. According to the results of the present study, nutrient uptake kinetics might differ among strains and this may be useful for optimizing temperature-dependent quantitative removal of nitrogen from water column in land-based cultivation. This suggests a clear future requirement to ascertain the nutrient-uptake kinetics of each strain of asexual variant.

For practicing land-based cultivation on an industrial scale, it is impractical (and uneconomic) to have to adjust the seawater temperature in the tank by external means. Therefore, seawater pumped from offshore or from saline wells, with seasonally fluctuating temperatures, needs to be used as it is for land-based cultivation. In order to improve and maximize productivity for such seasonal changes in seawater temperature, information about growth and sporulation responses of asexual variants is required, as in the present study. 

Currently, several land-based cultivation facilities in southern Japan are facing decreases in productivity due to reproductive maturation and pausing of growth at 20 °C or higher in summer. The use of cultivars with where sporulation does not occur until much higher temperatures, such as Strain 6 in the present study, would be one means to enable stable year-round cultivation in these southern areas. In contrast, higher productivity in the low winter temperature of northern areas require strains with higher growth rates at such temperatures. From the results of the present study, the growth rates of Strains 1 and 3 were significantly higher than other strains at 10 °C and 20 °C, respectively, so these strains can be regarded as elite cultivars at those temperatures. 

It might also be effective to use these cultivars seasonally according to observed changes in seawater temperature. The present study revealed that asexual variants of *U. prolifera* cover a wide phenotypic range of thermosensitivity as a result of natural selection. With regard to preserving valuable characteristics as an elite cultivar, selection from asexual variants is a useful technique, because in sexual strains the occurrence of recombination may result in the loss of the required optimal responses. Therefore, evaluating and utilizing of these asexual variants as a resource pool of candidates for elite cultivars will help to support optimization of productivity and expand the cultivable area of algae such as *U. prolifera*.

## 4. Materials and Methods

### 4.1. Collection and Stock Maintenance of Thalli

*Ulva prolifera* thalli were collected from the estuaries of six Japanese rivers (see Table 1). To confirm the life cycle of all thalli collected at each locality, sporulation and releasing zoids were conducted according to Hiraoka et al. (2003) [6]. Zoids of samples from sites 1 to 4 and 6 were found to be biflagellate. These thalli were confirmed as asexual variants, since their zoids showed negative phototaxis and were obviously bigger than both male and female gametes reported in previous studies [6]. Zoids of site 5 were quadriflagellate. Thalli cultured from the quadriflagellate site 5 zoids released the same type of quadriflagellate zoids again. More than two generations were repeated and all released quadriflagellate zoids, confirming that site 5 thalli were obligate asexual variants. A unialgal culture strain was established for each locality (Table 1, Strains 1–6) at Usa Marine Biological Institute, Kochi University. All strains were transported to the Yuriage Factory, Riken Food Co., Ltd., in Natori City, Miyagi Prefecture, and their seeding stocks for the growth and maturation experiments were prepared according to the GC method for unattached (free-floating) macroalgal culture [13]. Thallus clusters were produced according to the method of Hiraoka et al. (2020) with slight modifications [21]. Synchronous zoid formation in each strain was induced by cutting a well-developed thallus into small fragments of 1–2 mm in length, which were washed in sterilized fresh water for approximately 10 s and cultured in a Petri dish containing 40 mL Enriched Seawater (ES) medium [31] at 20 °C under a 12 h:12 h L:D cycle, with light of 150 µmol photons m^−2^ s^−1^. Under these conditions, thallus fragments released biflagellate (Strains 1–4 and 6) or quadriflagellate (Strain 5) zoids within 3 days.

Aliquots of zoid suspension densely concentrated using their phototactic response were placed in Petri dishes, adjusted to a density of >10^4^ zoids per 1 mL medium, and incubated under the same conditions as above. After 3 weeks, germlings grew at a high density on the bottom of the dish and attached to one another to form aggregations with the appearance of a green mat. The aggregations were scraped off the dish without harming them, torn into numerous small clusters and cultured with gentle aeration, allowing to drift freely within the vessel. When they attained a length of 1 mm or more, they were statically stocked under weak light (<50 µmol photons m^−2^ s^−1^ under 12 h:12 h L:D cycle) at 20 °C for one week until used for growth-rate and sporulation-response experiments.

### 4.2. Molecular Analysis

Total genomic DNA was extracted from a small fragment of living material using BT Chelex^®^ 100 Resin (Bio-Rad Cat# 143-2832, Hercules, CA, USA). A fragment of approximately 1 cm of each sample was ground in a 2 mL tube with 100 μL of 10% Chelex solution using a homogenization pestle at room temperature and incubated at 95 °C for 20 min, shaken at the middle and end of the 20 min period. The mixture was then cooled and centrifuged at 4000 rpm for 2 min.

Part of the sixth exon of the *hsp90* gene sequence was amplified using the primer pair of *hsp90*-6F (5′-GCAGACCCAGAAAGTGATCTATTAYATCA-3′) and *hsp90*-6R (5′-GCAGGYTCATCCAGACTAAATCC-3′), as reported by Ogawa et al. (2014) [9]. PCR amplifications were carried out using KOD FX Neo (ToYoBo, Osaka, Japan) and performed using a thermal cycler for 35 cycles of denaturation at 98 °C for 10 s, annealing at 55 °C for 30 s, and extension at 68 °C for 30 s, followed by a final hold at 10 °C. PCR products were sequenced by Fasmac (Atsugi, Kanagawa, Japan).

### 4.3. Growth Rate at Different Temperatures 

To reduce the lag phase growth of the stocked materials, hundreds of thallus clusters for each strain were pre-cultured in a round 3L-flask with continuous aeration for 7 days. The flask was filled with sterilized seawater containing half-strength ES medium [31]. Temperature and light conditions for growth of germlings were as described above. The medium was changed every day. When the thallus clusters grew to 5–10 mm in length in this pre-culture, 8–12 clusters (0.01 g-wet, Figure 6a) were transferred to 500 mL-flasks and cultured with aeration at 10, 15, 20, 25, 30 °C for 8 days (Figure 6b). Light was provided from an LED unit (3LH-64, NK System Co., Ltd. Osaka, Japan) at 150 µmol photons m^−2^s^−1^ with a 12 h:12 h L:D cycle in the incubator (CN-40A, Mitsubishi Electric Engineering Co., Ltd., Tokyo, Japan). Half-strength ES medium was used as culture medium and changed every day. The temperature range and light intensity were set according to the previous study about the RGR of this alga vs. abiotic conditions [21], the RGR was saturated at a light intensity of >67 µmol photons m^−2^ s^−1^ and indicated broad ranges from 10 °C to 30 °C. The pre-culture experiment decided the enrichment medium condition; we confirmed that the half-strength of ES medium could be sufficient for the RGR by changing it daily. To determine the fresh mass of living material without causing damage by drying, thallus clusters were held between sterilized paper towels four times to carefully remove water on the surface, immediately placed in a Petri dish (9 cm in diameter) filled with half-strength ES medium on an electronic balance (0.1 mg accuracy), quantified, and returned to the same culture condition. This mass measurement was made within a few minutes at the end of the light period every day, equally spaced at 24 h intervals. Relative growth rate (RGR: the continuously accelerating growth of algae during the exponential phase; Appendix A) was calculated using the following equation:RGR = (lnW_1_ − lnW_0_) day^−1^
where W_0_ is the initial fresh mass in the culture at time zero, and W_1_ is the mass after 24 h.

### 4.4. Sporulation at Different Temperatures

Between five and eight precultured thallus clusters were selected and separated individually in the center part of clusters being careful not to injure the thallus and interfere with the release the sporulation inhibitor [32]. Three intact thalli (length 1 mm) from each strain were selected and each individual placed in a separate 500 mL flask and cultured at 10, 15, 20, 25, 30 °C under 150 µmol photons m^−2^s^−1^ with a 12 h:12 h L:D cycle for 12 days, using the same incubators and LED units as for the growth rate experiments. All were incubated in half-strength ES medium renewed every 2 days. When the medium was changed, the thalli were placed in a 9 cm Petri dish filled with half-strength ES medium and the thallus surface was observed by light microscopy for the presence or absence of sporulation (Figure 6c–e) and, using a digital camera attachment, recorded as digital images.

### 4.5. Carbon and Nitrogen Contents at Different Temperatures

For all strains used in the growth rate experiment at different temperatures, five clusters were randomly selected after final measurements had been taken. Since the light intensity used was above the compensation irradiance for photosynthesis [23], and the culture medium (half-strength ES; approximately 420 µM as nitrate) was changed every day, the thalli were considered to be supplied with sufficient carbon and nitrogen for normal growth to occur. Seawater was carefully blotted from the thallus surface of the cluster samples, which were placed in a dry oven (EYELA WFO-500, Tokyo Rikakikai Co., Ltd., Tokyo, Japan) for 12 h at 90 °C. The dried thalli were then each pulverized with a pestle and mortar and carbon and nitrogen content were measured using a CHN analyzer (Flash 2000, Thermofisher Scientific, Waltham, MA, USA).

### 4.6. Statistical Analysis

All data are presented as mean ± S.E. Significant differences in RGR, carbon content, and nitrogen content among different cultivation temperatures and different strains were identified by the Kruskal–Wallis test followed by Steel–Dwass multiple comparison tests. A nonparametric procedure was chosen because not all of the data were normally distributed or homoscedastic. 

## Figures and Tables

**Figure 1 plants-10-02256-f001:**
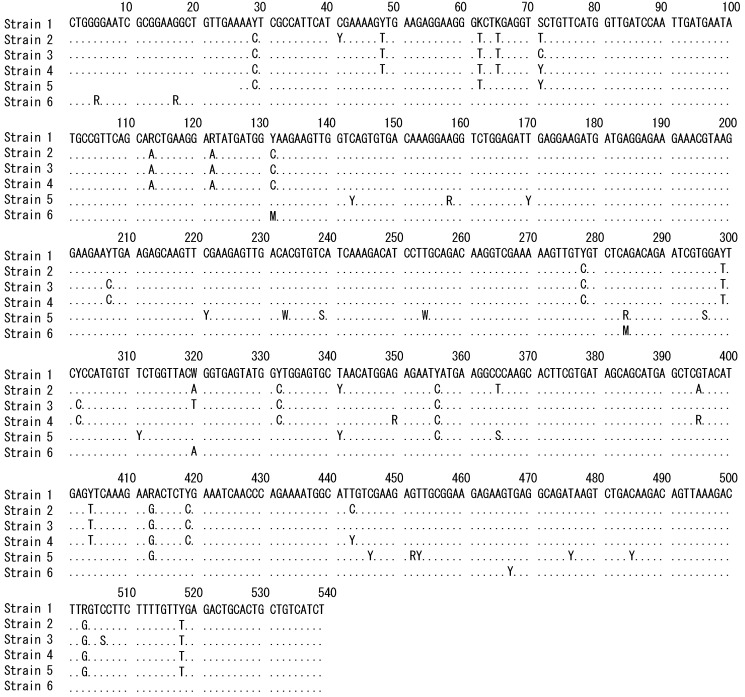
Comparison of *hsp90* gene sequences from *Ulva prolifera* thalli (undergoing an asexual life cycle) collected from six Japanese strains (see Table 1). Dots indicate identity with Strain 1; blanks indicate deletions; and double rows of dots and the following characters indicate the presence of different alleles: K, either G or T; M, either A or C; R, either A or G; S, either C or G; W, either A or T; Y, either C or T.

**Figure 2 plants-10-02256-f002:**
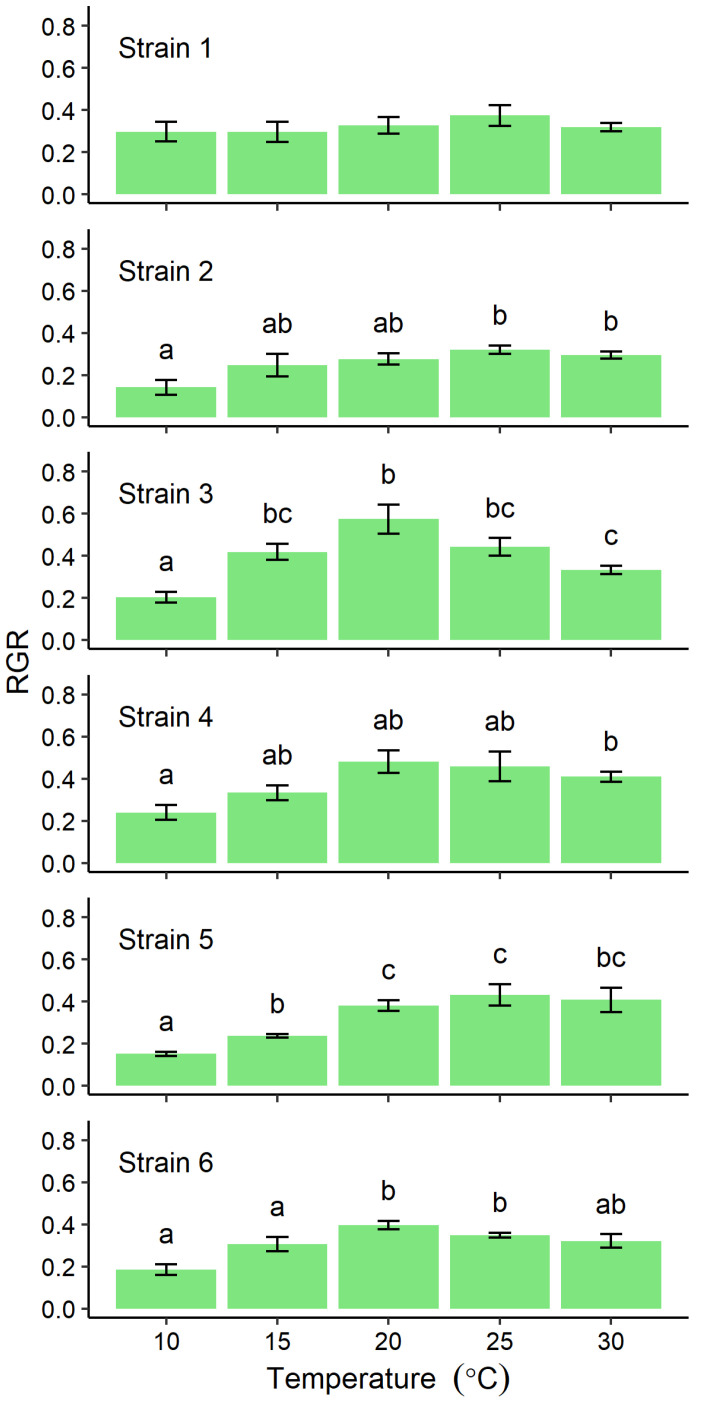
Relative growth rate (RGR) of *Ulva prolifera* thalli from each of six localities incubated in vitro at one of five different temperatures. The RGRs (*n* = 5) were calculated from four consecutive samples linearly arranged between 0.01 and 0.1 g (Appendix A). Error bars indicate standard error of the mean. Different lowercase letters indicate significant differences (*p* < 0.05) among different temperatures. The results of statistical analysis among strains at each cultivation temperature were shown in Appendix A.

**Figure 3 plants-10-02256-f003:**
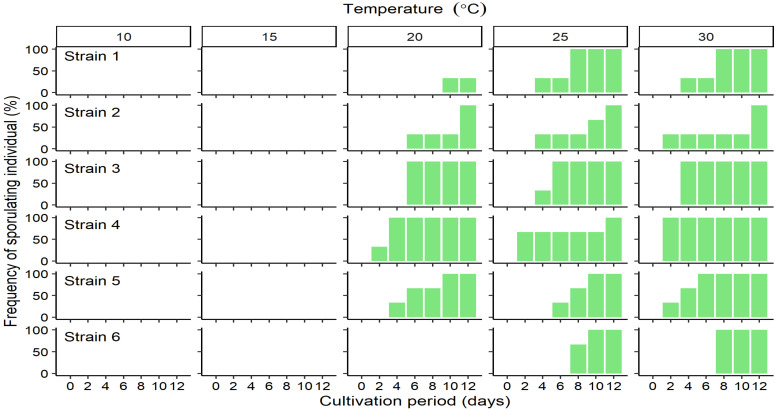
Changes in the frequency of occurrence of sporulating individuals of *Ulva prolifera* incubated in vitro at one of five different temperatures. Values are means ± standard error; *n* = 5 individuals.

**Figure 4 plants-10-02256-f004:**
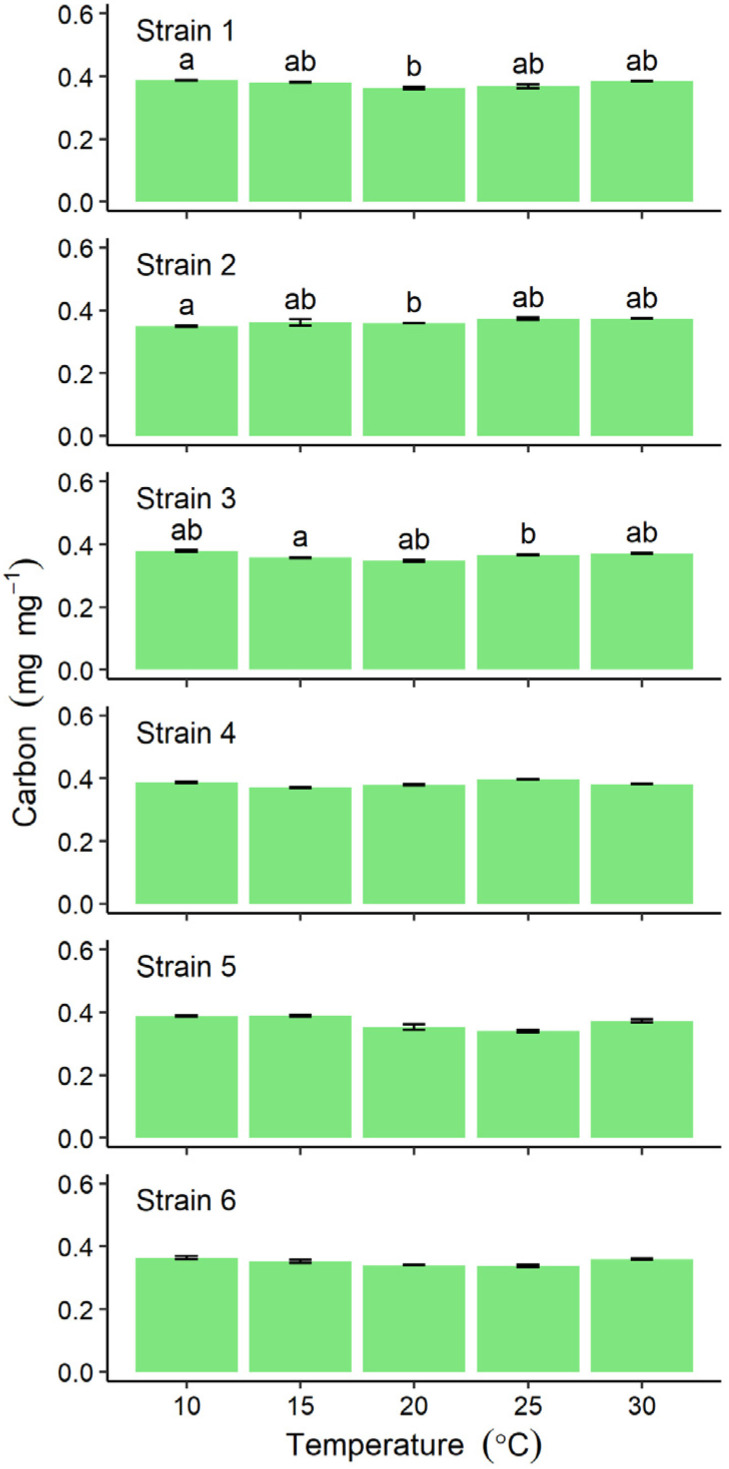
Comparison of carbon content in six strains of *Ulva prolifera* incubated in vitro at one of five different temperatures. Values are means ± standard error; *n* = 5 individuals. Different lower-case letters indicate significant differences (*p* < 0.05) among different temperatures. The results of statistical analysis among strains at each cultivation temperature were shown in Appendix A.

**Figure 5 plants-10-02256-f005:**
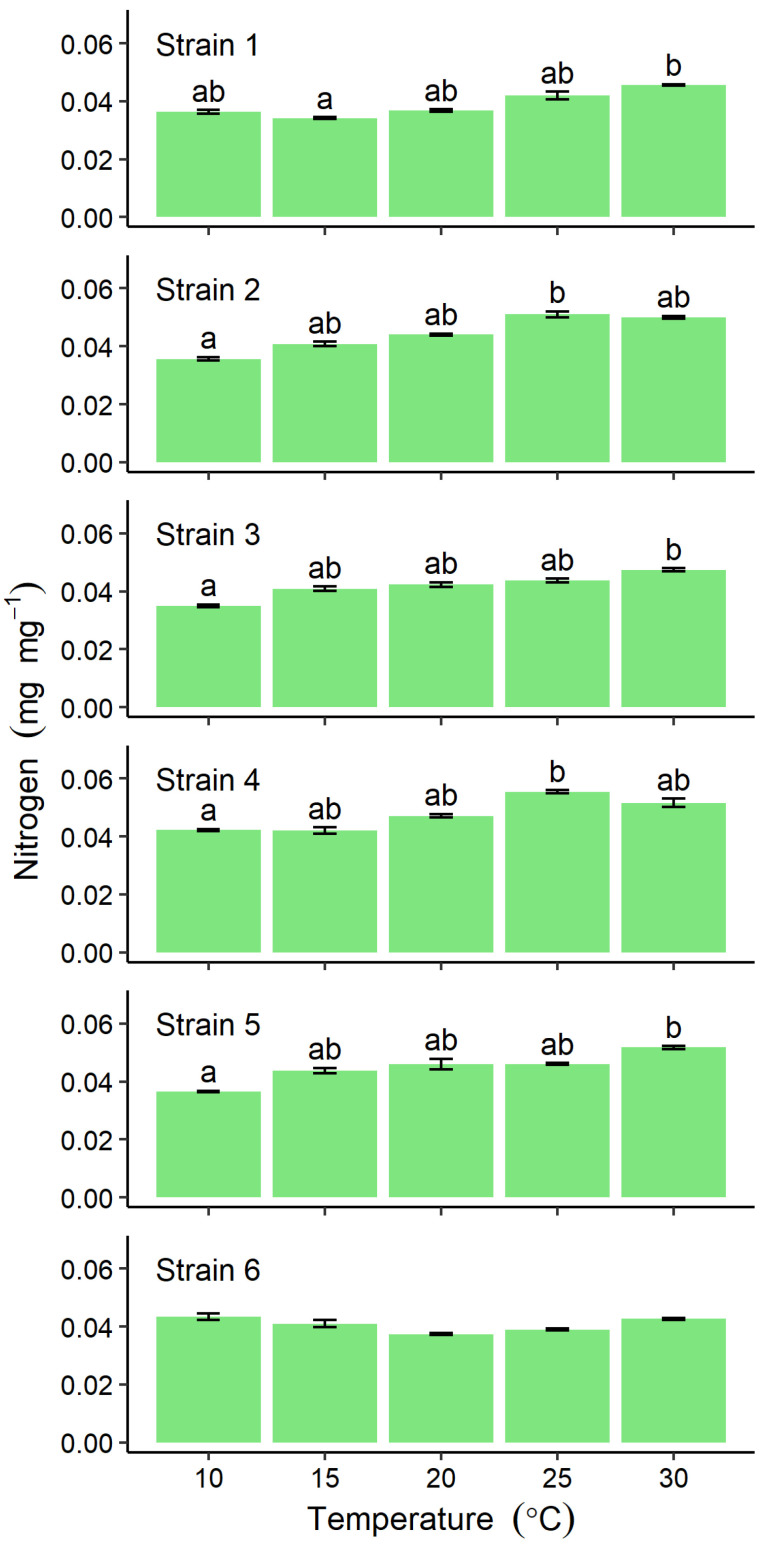
Comparison of nitrogen content in six strains of *Ulva prolifera* incubated in vitro at one of five different temperatures. Values are means ± standard error; *n* = 5 individuals. Different lowercase letters indicate significant differences (*p* < 0.05) among different temperatures. The results of statistical analysis among strains at each cultivation temperature were shown in Appendix A.

**Figure 6 plants-10-02256-f006:**
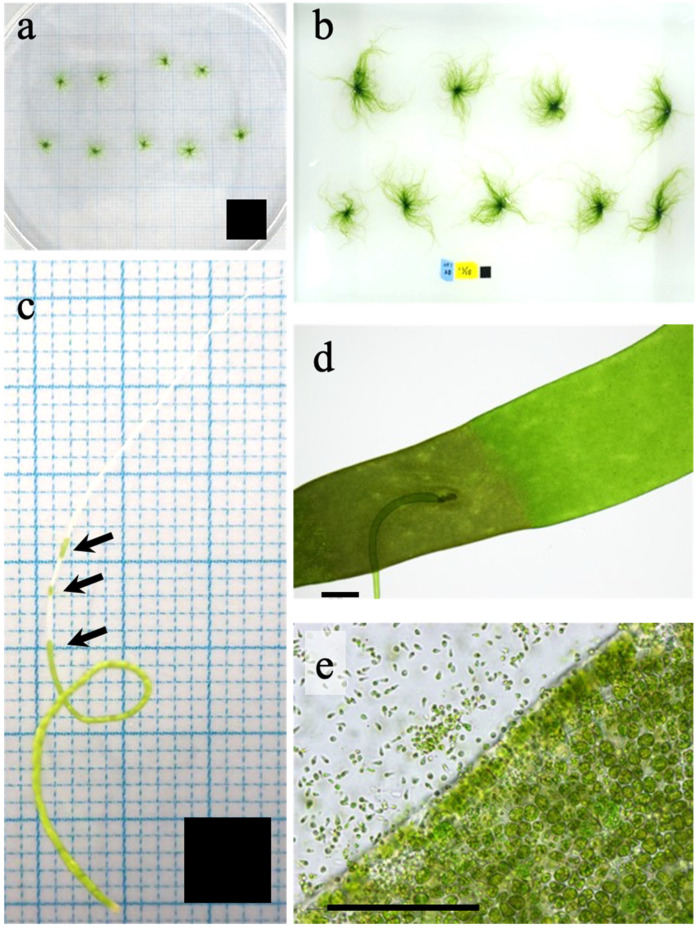
Representative images of *Ulva prolifera* used for the present study. (**a**,**b**) Thalli produced by the germling cluster method for the growth experiment: (**a**) initial thalli; (**b**) thalli after 8 d of cultivation. (**c**–**e**) Thallus for sporulation experiment: (**c**) sporulating individual with sporulated cells (arrows), (**d**) dark coloration indicating formation of sporulated cells, and (**e**) zoids released from sporulating part of thallus. Solid squares indicate 1 cm^2^ (**a**,**b**) and 0.25 cm^2^ (**c**); bars (**d**,**e**), 300 µm.

**Table 1 plants-10-02256-t001:** Basic information on the *Ulva prolifera* strains collected from six estuarine localities in Japan for use in this study.

River	City or Town, Prefecture or Subprefecture	Strain No.	Latitude and Longitude
Oboro	Akkeshi, Kushiro	1	43°04′33.7″ N, 144°50′16.6″ E
Sekiguchi	Yamada, Iwate	2	39°28′28.5″ N, 141°57′03.3″ E
Orikasa	Yamada, Iwate	3	39°26′57.4″ N, 141°57′43.5″ E
Sakari	Ofunato, Iwate	4	39°04′50.0″ N, 141°43′10.0″ E
Natori	Natori, Miyagi	5	38°10′49.6″ N, 140°56′51.7″ E
Takeshima	Shimanto, Kochi	6	32°57′44.5″ N, 132°58′34.0″ E

The Oboro River is on Hokkaido Island, the Takeshima River is on Shikoku Island, and the remaining rivers are in northeastern Honshu.

## Data Availability

Data is contained within the article and Appendix A.

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
