# Peer review of "Different Growth and Sporulation Responses to Temperature Gradient among Obligate Apomictic Strains of Ulva prolifera"

_plants, 2021, doi:10.3390/plants10112256_

Round 1

Reviewer 1 Report

I reviewed the manuscript entitled “Different Growth and Sporulation Responses to Temperature Gradient among Obligate Apomictic Strains of Ulva prolifera” by Yoichi Sato and colleagues.

The topic is really relevant and current, the study was conducted in a very rigorous manner and is presented in a clear and concise manner. The results obtained are well presented and discussed and represent an important contribution to research in the field of macroalgae cultivation with important implications for their cultivation on an industrial scale.

I think that the manuscript undoubtedly deserves to be published.

I report below only a few small questions and suggestions for each ms section:

INTRODUCTION: In the aim of the study authors may want to mention that the study should be useful in an industrial aquaculture perspective.

RESULTS & MATERIAL and METHODS: how difference among strains were investigated? From M&M section and figures I understand that the Kruskall Wallis tests were performed for testing differences among temperatures, separately for each strain. It is correct? Or also differences among strains were tested? Please, add details about statistical tests, indicating clearly which factor was tested and how many test were performed. Please note that if differences among strains have not been statistically tested, significant differences among strains should not be claimed.

In M&M maybe some additional info about the choice of the range of temperature should be added. And also about the choice of light intensity and half-strength medium.

FIGURES: Authors should consider to present data in Fig. 2, 4, 5 as boxplot, showing median values, instead of bar plot, given that they performed a statistical test (Kruskal Wallis) comparing median values. It is only a suggestion.

Please, note that in Fig.2 caption Ulva prolifera is not in italics.

DISCUSSION

Line 162: delete characteristics (repetition)

Line 183: how correlation of carbon content and RGR has been tested? Data are not shown?

Author Response

YS: Thank you for careful and constructive comments. As your suggestions, we modified and improved the manuscripts. In the manuscript, the revised sentences, as suggested, are highlighted in yellow. Answers to specific comments are written below.

INTRODUCTION: In the aim of the study authors may want to mention that the study should be useful in an industrial aquaculture perspective.

YS: Thank you for your carefully comment. Yes, we aimed to not only understanding the physiological characteristics for growth and sporulation responses of asexual variants of U. prolifera among recognized strains, but also applying the information to an industrial aquaculture field. Therefore, we added the sentence to the end of Introduction as follows;

[L88]... to apply the industrial aquaculture perspective.

RESULTS & MATERIAL and METHODS: how difference among strains were investigated? From M&M section and figures I understand that the Kruskall Wallis tests were performed for testing differences among temperatures, separately for each strain. It is correct? Or also differences among strains were tested? Please, add details about statistical tests, indicating clearly which factor was tested and how many test were performed. Please note that if differences among strains have not been statistically tested, significant differences among strains should not be claimed.

YS: Thank you for your kind comment. Yes, the Kruskal Wallis test were performed among strains. We added the sentence the information into the 4.6 of M.M [L.365] and caption of Figure 2, 4, and 5. We described the statistical results among strains in supplemental table.

In M&M maybe some additional info about the choice of the range of temperature should be added. And also about the choice of light intensity and half-strength medium.

YS: The previous study [21] revealed the environmental conditions for the growth of U. prolifera as follows; 1) since light intensities of >67 µmol photons m-2 s-1 were saturated for growth, they used 100–200µmol photons m-2 s-1 for the standard light condition for growth experiment, 2) RGR of this alga indicated a broad range of temperature of 10–30ºC. Concerning the strength of the enrichment medium, we confirmed beforehand that the RGR of this alga could maintain the level of ES medium even though half-strength ED medium. We added the information in MM as follows [L.305–310];

The temperature range and light intensity were set according to the previous study about the RGR of this alga vs. abiotic conditions; the RGR was saturated at a light intensity of >67 µmol photons m-2 s-1, 2) and indicated broad ranges from 10 ºC to 30 ºC. The pre-culture experiment decided the enrichment medium condition; we confirmed that the half-strength of ES medium could be sufficient for the RGR by changing it daily.

FIGURES: Authors should consider to present data in Fig. 2, 4, 5 as boxplot, showing median values, instead of bar plot, given that they performed a statistical test (Kruskal Wallis) comparing median values. It is only a suggestion.
YS: Thank you for your suggestion and I can accept your opinion; however, we would like to focus to the comparison of average values and use the bar plot graph.

Please, note that in Fig.2 caption Ulva prolifera is not in italics.
YS: [Fig.2] Revised.

DISCUSSION

Line 162: delete characteristics (repetition)
YS: [L.173] Deleted the “characteristics”.

Line 183: how correlation of carbon content and RGR has been tested? Data are not shown?
YS: [L.194] Thank you for carefully comment. Since the sentence were unclear, we deleted it.

Reviewer 2 Report

The authors characterized a green alga, Ulva, with regard to its sporulation patterns in different temperatures. The study was well organized and clearly presented. I recommend it for publication. 

The study was well organized but severely limited in scope. It is a simple study; interesting, yet leaves much to be desired in the way of sample size and data output. For example, the authors take genotypes from a single gene, but whole genomes can easily be sequenced for little more than the cost of PCR reagents. In short, the authors should use more cultivars and perform WGS to increase the potential impact of their study.

Author Response

We appreciate your positive comments and recommendation to publish.
We accepted other reviewers' comments and revised the manuscript.

Reviewer 3 Report

This is an interesting research work on the growth of Ulva prolifera. The impact of this work is very important, considering it is a commercially cultivated species. The authors prove that different cultivars can (and should) be used in distinct areas of Japan with different conditions (mostly water temperature) to improve productivity.

The paper is very well written, the introduction provides enough information to understand the subject.

The methods are standard and well described, although I have some questions regarding the methods used.

The results are clearly presented, the discussion is clear, and states other studies addressing the subject. It presents enough information to draw clear conclusions, supported by the results.

Therefore, I believe that the paper deserves to be published, after minor amendments that are indicated below to improve the document.

Introduction 

Line 42 –  So, just to make it clear, both biflagellate and tetraflagellate zoospores are diploid and produced by the sporophytes? What is the ecological significance of these two types of cells? Is it known? Maybe the authors could further explore this issue.

Results and discussion

Line 94 – can the authors elaborate on this? I can see that the thalli are from different strains because they show slightly different sequences, but with the data shown I can’t see that they are heterozygous and diploid (sporophytes).

Line 109 – consider using the word higher instead of faster.

Line 170, 194, 234, 246 – the [#] number of the reference is missing.

Line 214 – delete the word “with”

Material and methods:

Line 236 – what about thalli 2 and 3? How did the authors test these thalli? What type of zoids did they produce? In line 251 the authors mention they are biflagellate.

Line 257 – how did the authors scrap the new aggregates, without harming them?

Line 298 – the authors weighed the thalli inside a Petri dish containing ES medium? Why the ES medium after drying with paper towels? How do they confirm the exact amount of ES medium in each Petri dish?

Figure 6 e) – when are there gametes produced through sporulation? I believe the authors mean spores/zoids?

Author Response

YS: Thank you for your constructive comments. All comments were acceptable and reflected to revised manuscript. We modified as follows. In the manuscript, the revised sentences, as suggested, are highlighted in yellow. Answers to specific comments are written below.

Introduction 

Line 42 –  So, just to make it clear, both biflagellate and tetraflagellate zoospores are diploid and produced by the sporophytes? What is the ecological significance of these two types of cells? Is it known? Maybe the authors could further explore this issue.
YS: No, both biflagellate and quadriflagellate zoospores are diploid and produced by the asexual variants, these zoospores are not produced by sporophytes. Sporophytes are generation producing meiospore at lifecycle with sexual reproduction, therefore, asexual variants producing both biflagellate or quadriflagellate diploid zoids are not sporophytes.

We added a few ecological information about these zoospores as follows;

[L.42–48]... including U. prolifera, are known to have two types of obligate asexual life cycles without sexual reproduction via meiosis and conjugation, reproducing through biflagellate or quadriflagellate diploid zoids specialized for asexual development, these zoids have negative phototaxis [3]. These asexual zoids were termed “zoosporoids” [4,5]. The quadriflagellate zoosporoids of obligate asexual life history were a length of <10µm; these were smaller in size than quadriflagellate meiospores of sexual life history (>11 µm length). On the other hand, biflagellate zoosporoids of obligate asexual life history were a length of 8–9 µm; these were distinguished from biflagellate games (6–7 µm length) [6].

Results and discussion

Line 94 – can the authors elaborate on this? I can see that the thalli are from different strains because they show slightly different sequences, but with the data shown I can’t see that they are heterozygous and diploid (sporophytes).
YS: We mentioned in MM as follows why these thalli were recognized to be diploid as follows, and previous studies have reported that the asexual strains of U. prolifera is diploid [7.8]. Alternative bases at some positions were evidence as all strains having heterozygous (detail in figure 1). hsp90 sequences are considered useful markers for estimating the genetic diversity and structure among within populations of Ulva. figure 1 indicates that all strains are heterozygous. For interpretation of the results, we referred to the reference [9].

[L.247–252]...These thalli were confirmed as asexual variants, since their zoids showed negative phototaxis and were obviously bigger than both male and female gametes reported in previous studies [6]. Zoids of type 5 were quadriflagellate. Thalli cultured from the quadriflagellate type 5 zoids released the same type of quadriflagellate zoids again. More than two generations were repeated and all released quadriflagellate zoids, confirming that type 5 thalli were obligate asexual variants.

Line 109 – consider using the word higher instead of faster.
YS: [L.114] Revised.

Line 170, 194, 234, 246 – the [#] number of the reference is missing.
YS: As previous published paper of Plants, the references cited at the beginning of the sentence were not numbered and I followed them. I ask to editorial office.

Line 214 – delete the word “with”
YS: Revised as follows.
[L.228–230] In contrast, higher productivity in the low winter temperature of northern areas requires strains with higher growth rates at such temperature.

Material and methods:

Line 236 – what about thalli 2 and 3? How did the authors test these thalli? What type of zoids did they produce? In line 251 the authors mention they are biflagellate.
YS: Thalli 1,2,3,4 and 6 were confirmed as asexual variants, since their zoids showed negative photo-taxis and were obviously bigger than both male and female gametes reported in previous studies [6].

Line 257 – how did the authors scrap the new aggregates, without harming them?
YS: Yes, I carefully did the scrap without harming them by using a spatula of TeflonTM. We added the sentence as follows, thank you for careful comment.

[L. 269–270] The aggregations were scraped off the dish without harming them, torn into...

Line 298 – the authors weighed the thalli inside a Petri dish containing ES medium? Why the ES medium after drying with paper towels? How do they confirm the exact amount of ES medium in each Petri dish?
YS: Since the petri dish containing the medium was tare in advance and the thalli were blotted by paper towel to remove moisture , we were able to accurately measure the weight of the thalli.

Figure 6 e) – when are there gametes produced through sporulation? I believe the authors mean spores/zoids?
YS: Zoid formation in each strain was induced by cutting a well-developed thallus into small fragments of 1–2 mm in length. We revised form “gametes” to “zoids”